# Enhanced Thermochromic Performance of VO_2_ Nanoparticles by Quenching Process

**DOI:** 10.3390/nano13152252

**Published:** 2023-08-04

**Authors:** Senwei Wu, Longxiao Zhou, Bin Li, Shouqin Tian, Xiujian Zhao

**Affiliations:** State Key Laboratory of Silicate Materials for Architectures, Wuhan University of Technology (WUT), No. 122, Luoshi Road, Wuhan 430070, China; senwei_wu@whut.edu.cn (S.W.); zhoulongxiao@whut.edu.cn (L.Z.); libin625@whut.edu.cn (B.L.)

**Keywords:** VO_2_, quenching process, dispersion structure, thermochromic properties

## Abstract

Vanadium dioxide (VO_2_) has been a promising energy-saving material due to its reversible metal-insulator transition (MIT) performance. However, the application of VO_2_ films has been seriously restricted due to the intrinsic low solar-energy modulation ability (Δ*T*_sol_) and low luminous transmittance (*T*_lum_) of VO_2_. In order to solve the problems, the surface structure of VO_2_ particles was regulated by the quenching process and the VO_2_ dispersed films were fabricated by spin coating. Characterizations showed that the VO_2_ particles quenched in deionized water or ethanolreserved VO_2_(M) phase structure and they were accompanied by surface lattice distortion compared to the pristine VO_2_. Such distortion structure contributed to less aggregation and highly individual dispersion of the quenched particles in nanocomposite films. The corresponding film of VO_2_ quenched in water exhibited much higher Δ*T*_sol_ with an increment of 42.5% from 8.8% of the original VO_2_ film, because of the significant localized surface plasmon resonance (LSPR) effect. The film fabricated from the VO_2_ quenched in ethanol presented enhanced thermochromic properties with 15.2% of Δ*T*_sol_ and 62.5% of *T*_lum_. It was found that the excellent *T*_lum_ resulted from the highly uniform dispersion state of the quenched VO_2_ nanoparticles. In summary, the study provided a facile way to fabricate well-dispersed VO_2_ nanocomposite films and to facilitate the industrialization development of VO_2_ thermochromic films in the smart window field.

## 1. Introduction

Since Morin F. J. discovered the reversible metal-insulator transition (MIT) behavior of vanadium oxides in 1959 [1], VO, V_2_O_3_, VO_2_, V_6_O_13_, V_3_O_7_, and V_2_O_5_ [1,2,3] have been reported undergoing MIT at different ambient temperatures and showing significant gaps of the electrical, optical, magnetic properties before and after the MIT. Among them, the phase transition temperature (*T*_c_) of VO_2_, 341 K (68 °C), is the closest to room temperature, which makes VO_2_ a promising candidate for intelligent film for smart windows. Below *T*_c_, the strong electron-correlated VO_2_ is monoclinic (M phase, P2_1_/c) and highly transparent to NIR, which helps warm up the indoor room. When the ambient temperature rises above *T*_c_, the VO_2_ crystal transits to be tetragonal (R phase, P4_2_/mnm) and reflective to the NIR, which is beneficial for blocking out the thermal radiation and cooling down the room temperature. Such characteristic of temperature-sensitive response of VO_2_ is expected to enable intelligent regulation of indoor temperature, thereby reducing the energy consumption of architectures [2,3,4].

However, challenges remain in balancing the admirable luminous transmittance (*T*_lum_) for illuminance and the appreciable solar-energy modulation ability (Δ*T*_sol_) for temperature regulations [5,6,7]. Such problems severely hinder the application of VO_2_ thermochromic smart windows. Up till now, plenty of strategies have been dedicated to the performance optimization of VO_2_. Element doping is confirmed to be an effective way to diminish the yellow color and boost the visible transmittance *T*_lum_ [3,5,8], and F [9], Mg [9,10], Ti [11], and Zr [12] are common dopants. Besides, VO_2_ nano-/microstructure engineering [4], such as porous structure [13,14], grid structure [15], and biomimetic patterning [16] of VO_2_ film, has been testified to cater for the optimization of thermochromic properties. In addition, multilayer construction [17,18,19] based on VO_2_ film becomes an effective approach to enhance thermochromic properties, including but not limited to oxide buffer layers [20], antireflection layers [21], and noble metal layers [22]. Apart from multilayer structure, VO_2_ nanocomposite coating by means of distributing VO_2_ particles into matrix materials is another feasible way to improve *T*_lum_ and Δ*T*_sol_. When the ambient temperature is above *T*_c_, the VO_2_(R) nanoparticles, with grain size much smaller than the wavelength of the incident light, present the localized surface plasma resonance (LSPR) effect around 1200 nm. Such an effect enables to enlarge the absorption and decreased transmittance of the infrared light. On the other hand, VO_2_(M) nanoparticles below *T*_c_ do not possess the characteristic of LSPR and the corresponding films are highly transparent to NIR. Such a NIR transmittance gap between VO_2_(M) and VO_2_(R) results in much higher Δ*T*_sol_. Additionally, the transparent matrix covering isolated particles gives rise to higher *T*_lum_ of the VO_2_ nanocomposite coatings. Typically, SiO_2_ [23], SnO_2_ [24], TiO_2_ [25], and Si-Al oxide [26] are common inorganic matrixes for VO_2_ nanoparticles. Unfortunately, these matrixes require a strict annealing atmosphere or precisely uniform dispersion of VO_2_ in the nanocomposite films. Compared to inorganic matrix, organic agents, such as polyvinyl pyrrolidone (PVP) [13,27], polyurethane (PU) [28], polyvinyl butyral (PVB) [29], and acrylic resin [30], have become more favorable hosts due to their low refractive index and facile fabrication process. Gao et al. [31] synthesized a flexible coating by dispersing SiO_2_ capsuled monocrystal VO_2_ with good crystallinity, fine grain size (VO_2_@SiO_2_) into PU, which showed good optical properties (*T*_lum_ = 55.3%, Δ*T*_sol_ =7.5%). In addition, they further developed a roll-coated VO_2_ nanocomposite coating with smoothness, uniformity, and high transparency on 1200 × 1000 mm^2^ large-scale glass. By modulating the viscosity of the host PU, Zou et al. [32] upgraded the solar modulation ability of VO_2_/PU composite film from 6.6% to 14.5%, whereas the luminous transmittance was maintained 54%. On the other hand, PVP was widely reported to be an alternative matrix. Zhao et al. [13] mixed VO_2_ nanoparticles, ZnO nanoparticles, and PVP to fabricate a composite film, successfully enhancing *T*_lum_ from 54.9% to 63.9% and improving Δ*T*_sol_ from 9.9% to 11.3%. Recently, Tian et al. [33] demonstrated an in-situ synthesis route to coat VO_2_ nanocomposite on the glass surface by PVP decomposition and a mass transfer process during annealing, achieving remarkable optical performance with *T*_lum_ of 72.5% and Δ*T*_sol_ of 10.1%. Yet, there were rare reports about the interactions between the VO_2_ and the matrix.

In this work, high-purity VO_2_(M) particles obtained by a one-step annealing process were quenched to induce surface structural distortion. Deionized water and ethanol were used as the quenching solvents to modulate the surface structure of VO_2_ particles. As it turned out, the phase transition temperature of the quenched powders was slightly reduced by around 3 °C. In addition, the quenched VO_2_ nanoparticles were dispersed individually in the PVP when spin-coated to form the VO_2_ nanocomposite films. Such dispersion structures contributed to the optimization of the thermochromic properties of the film. It was discovered that the quenched VO_2_ particles showed high dispersity in the nanocomposite films and presented an evident LSPR effect, which contributed to the enhanced solar-energy modulation ability of the films. Moreover, the film fabricated by the ethanol-quenched VO_2_ uncovered excellent optical performance, accompanied by Δ*T*_sol_ of 15.2% and *T*_lum_ of 62.5%. The outstanding performance successfully achieved the simultaneous enhancement of both solar energy modulation ability and luminous transmittance of the film, and it was superior to most of the VO_2_ nanocomposite films in previous reports [21,23,27,31,32,33] that failed to make a positive balance of Δ*T*_sol_ and *T*_lum_. Herein, this work provided a facile quenching process to benefit the thermochromic performance of VO_2_ composite film and to drive its application in smart windows.

## 2. Methods

### 2.1. Fabrication of Quenched VO_2_(M) Nanoparticles

All reagents that were used directly were analytically pure and provided by Sinopharm Chemical Reagent Co., Ltd, Shanghai, China. A certain mass of vanadium pentoxide (V_2_O_5_, 0.8 g) powders and half the amount of ammonium bicarbonate (NH_4_HCO_3_, 0.4 g) particles were loaded together in a quartz crucible and then transferred into the tube furnace. The furnace was pumped below 50 Pa and heated around 550 °C until the powders changed to dark-blue VO_2_ particles. After that, the particles were processed with quenching treatment as Figure 1 described. In detail, VO_2_ particles were placed in a vacuum tube furnace for thermal insulation treatment for half an hour and then they were rapidly moved into the 0 °C quenching solution surrounded by an ice-water system. Extra ice bulks were constantly added to the system so that the quenching solvent was maintained at around 0 °C. Next, the beaker holding the above-mentioned solution mixed with quenched VO_2_ grains was put in a thermostatic oven to completely evaporate the solvent. Finally, the remaining nanoparticles in the beaker were collected as quenched VO_2_(M).

### 2.2. Fabrication of VO_2_-PVP Nanocomposite Film

The obtained VO_2_(M) nanoparticles were mixed with polyvinylpyrrolidone (PVP, K30) and ethanol in the mill tank. Intermittent ball milling gave rise to particles with smaller sizes to achieve thorough dispersion in the ethanol. Afterward, the mixture liquid was transferred into a centrifuge tube for solid-liquid separation. The upper liquid in the tube remained turbid as VO_2_ with extreme fine sizes dispersed well in PVP, which dissolved well in ethanol. With the evaporation of liquid in the upper suspension, VO_2_-PVP nanocomposites were collected and then made into a coating slurry by mixing them with additional ethanol. Continuous stirring and ultrasonic vibration were implemented to make the slurry homogeneous. Then spin-coating was carried out to form a sol/wet film by dropping the solution on a common soda-lime-silica glass substrate. In the end, ethanol was removed by heating to form the VO_2_-PVP nanocomposite films.

### 2.3. Characterization

X-ray diffraction (XRD, D8DISCOVER, Bruker, Billerica, MA, USA) with Cu Kα (λ = 0.154056 nm) serving as the source of radiation, and 3 kW of the output power)was adopted to determine the phase structures of the powders over the 2θ between 10° and 80°. A differential scanning calorimeter (DSC, DSC8500,PerkinElmer, Waltham, MA, USA) was used to examine the phase transition temperature of the powders with the temperature ranging from 0 °C to 100 °C at the rate of 5 °C/min in the heating/cooling loop, as illustrated in Equation (1). *T*_c_ refers to the average phase transition temperature of VO_2_ particles. *T*_c,h_ and *T*_c,c_ correspond to the phase transition temperature peak of VO_2_ in the heating and cooling stages, respectively. A field emission scanning electron microscopy (FE-SEM, Zeiss Ultra Plus, Carl Zeiss CMP GmbH, Oberkochen, Germany) was used to observe the morphology of both the composite powders and films. X-ray photoelectron spectroscopy (XPS, ESCALAB 250Xi, ThermoFisher, Waltham, MA, USA) was utilized to determine the element composition and valence. FT−IR spectrum (Nicolet6700, ThermoFisher, Waltham, MA, USA) was served to identify the functional groups of the samples.
(1)$$T_{c}=(T_{c,h}+T_{c,c})/2$$

The thermochromic performance of the films was measured from 300 to 2500 nm by a UV–vis–NIR spectrophotometer (UV-3600) equipped with a temperature-controlling device. The transmittance of the films at 20 °C and 90 °C corresponded to VO_2_(M) and VO_2_(R), respectively. The integrated luminous transmittance (*T*_lum_, 380 nm ≤ λ ≤ 780 nm) and solar transmittance (*T*_sol_, 300 nm ≤ λ ≤ 2500 nm) can be calculated according to Equations (2) and (3).
(2)$$T_{\mathrm{lum}}=\frac{\int^{}\varphi_{\mathrm{lum}}\left(\lambda \right)T\left(\lambda \right)d\lambda }{\int^{}\varphi_{\mathrm{lum}}\left(\lambda \right)d\lambda }$$
(3)$$T_{\mathrm{sol}}=\frac{\int^{}\varphi_{\mathrm{sol}}\left(\lambda \right)T\left(\lambda \right)d\lambda }{\int^{}\varphi_{\mathrm{sol}}\left(\lambda \right)d\lambda }$$

In the equations, *T*(λ) represents the film transmittance of light at a certain wavelength (λ), *φ*_lum_(λ) is the standard luminous efficiency function for the photopic vision of human eyes [11], and *φ*_sol_(λ) is the solar irradiance spectrum for air mass 1.5 corresponding to the sun standing 37° above the horizon. Δ*T*_sol_ is the difference value of *T*_sol_ at 20 °C and 90 °C, as given in Equation (4).
(4)$$\Delta T_{\mathrm{sol}}=T_{\mathrm{sol}}\left(20\;{}^{\circ}C\right)-T_{\mathrm{sol}}\left(90\;{}^{\circ}C\right)$$

## 3. Results and Discussion

### 3.1. Structure of the VO_2_ Nanoparticles

VO_2_ powders were obtained from V_2_O_5_ by a facial annealing reduction reaction. As shown in the TG-DSC pattern (Figure 1a) of the mixture of V_2_O_5_ and NH_4_HCO_3_, NH_4_HCO_3_ kept decomposing to release NH_3_ at the beginning during the homogeneous heating period, illustrating the continuous mass loss of the raw materials. Till 183.5 °C, NH_4_HCO_3_ decomposed completely and the mass loss in this period was 8.28% in total. The endothermic peak at 226.2 °C indicated that the absorbed H_2_O was released, accounting for 3.29% of the reaction agents. With the rising temperature, the reducibility of NH_3_ contributed to changing the vanadium (V) in V_2_O_5_ to lower valence, accompanied by an exothermic process at 360.0 °C, and finally reached the stable phase until 400 °C, as no mass loss or well as energy exchange could be observed in the pattern. This result suggested that 400 °C was suitable for V_2_O_5_ reduction. Additionally, the small temperature deviation between DSC and DTG could be attributed to the errors caused by the test instrument. Figure 1b showed the XRD patterns of the VO_2_ samples annealed from V_2_O_5_ and NH_4_HCO_3_ mixture at 400 °C, 450 °C, 550 °C, and 550 °C. Although the main diffraction peaks (27.8°, 35.7°, 37.8°, and 55.8°) matched well with VO_2_(M) (PDF #44-0252), extra peaks with evident intensity (25.3°, 33.5°, and 49.5°) belonging to V_6_O_13_ (PDF #27-1318) could be observed for the samples obtained at 400 °C, which meant the incomplete reduction of V_2_O_5_ to V_6_O_13_. Such a result disagreed with the ideal reaction temperature in Figure 1a. The reason could be attributed to the fact that the furnace chamber in the annealing process was too large, and the effective heating interval of the furnace was small, which caused partial energy loss. When raising the annealing temperature to 450 °C, the fabricated samples came across the same situation. Interestingly, the diffraction peaks belonging to V_6_O_13_ of the VO_2_ sample reduced at 450 °C were much weaker than those of VO_2_ reduced at 400 °C, testifying that the proportion of V_6_O_13_ among the VO_2_ samples annealed at 450 °C was much less than that of the VO_2_ annealed at 400 °C. Thus, a higher annealing temperature was demanded to overcome the energy barrier for reducing V_2_O_5_ into VO_2_ thoroughly. In addition, it turned out that the material annealed at 500 °C became pure VO_2_(M) with sharp diffraction peaks. Such peaks implied that the 500 °C reduced VO_2_ featured excellent crystallinity. Comparably, the powders annealed at 550 °C remained pure VO_2_(M) structure but exhibited a decreased peak intensity, indicating worse crystallinity of the related VO_2_ powders. This might be caused by the unexpected grain agglomeration during the thermal insulation at higher temperatures. Therefore, 500 °C was selected as the ideal temperature to produce pristine VO_2_ (labeled as P-VO_2_).

To observe the metal-insulator transition, P-VO_2_ was characterized through in-situ XRD technique, and the results were shown in Figure 2a,b, and Figure 2b was the enlarged view of the pink area in Figure 2a. The VO_2_ stayed the monoclinic structure below 45 °C, and the main peak (27.76°) corresponded to the (011) crystal plane of VO_2_(M). When the temperature rose above 65 °C, the peak shifted to 27.63°, revealing that the VO_2_ crystals had transferred to the rutile phase, featured with the (110) crystal plane of VO_2_(R) (PDF #73-2362). This phenomenon suggested that the phase transition temperature of P-VO_2_ ranged between 45 °C and 65 °C. This was in agreement with previous reports [5] for VO_2_ particles. When the VO_2_ particles cooled down from 75 °C to 65 °C, VO_2_ crystals transited back to monoclinic structure from rutile structure. This helped obverse the evident and reversible MIT behavior between VO_2_(M) and VO_2_(R) in the previous report [34]. Furthermore, the phase transition temperature of the cooling stage was inconsistent with that of the heating stage, confirming the thermal hysteresis loop feature of VO_2_ in the phase transition stage. It was the MIT behavior that made VO_2_ a promising material for smart windows.

In order to cause surface distortion for the crystal, P-VO_2_ featured with the notable MIT behavior was placed in a 0 °C ice-water system for quenching treatment as Figure 1 illustrated. According to the difference in quenching solvent, the VO_2_ quenched in deionized water was named H-VO_2_ and the one quenched in ethanol was noted as E-VO_2_. The XRD pattern of VO_2_ under different quenching solvents was shown in Figure 3a. Compared to the pristine VO_2_ without the quenching process, both the H-VO_2_ and E-VO_2_ powders held the phase structure of VO_2_(M) in Figure 1b. A smaller peak located at 25.3° appeared in the quenched samples, which could be the localized oxidation caused by exposure to air during powder transfer. It is worth noting that quenched particles presented the VO_2_(M) phase and manifested great crystallinity. In addition, surface lattice distortion of the VO_2_ was assumed to occur after the quenching process [35], which would be discussed in the next part. According to the Scherrer equation, the average grain size of E-VO_2_ and H-VO_2_ were calculated as 30.08 nm and 17.06 nm, respectively, larger than the average size (12.11 nm) of P-VO_2_. To explain the mean grain size changes of the three samples, an enlarged view of the XRD diffraction angle ranging from 27.0° to 28.5° (Organe area in Figure 3a) was presented to illustrate the half-width changes of the peaks. As shown in Figure 3b, the diffraction curve of P-VO_2_ exhibited the greatest widening state compared to E-VO_2_ and H-VO_2_. Such a state endowed P-VO_2_ with the largest half-width and thus, the smallest grain size among the three particles. On the other hand, the half-width of the E-VO_2_ (011) crystal plane diffraction peak was approaching that of H-VO_2_. However, the diffraction peak of E-VO_2_ shifted to a higher position at 27.91°, compared to the peak at 27.86° of P-VO_2_. Such a shift to a higher angle caused the fact that the grain size of E-VO_2_ was a bit larger than that of H-VO_2_. This phenomenon might be attributed to the hydroxy group attached to the surface of VO_2_ when the heated particles encountered an ice-water/ice-ethanol system and thus, leading to surface lattice distortion of the crystals. Moreover, thermal insulation during the quenching process in Figure 1 was also beneficial to grain growth.

With the goal of further determining the effect of the quenching process on VO_2_, the XPS technique was used to identify the chemical environment of elements, and the results were presented in Figure 4. Regarding the C1s peak at 284.80 eV as the calibration position, the full spectrum (Figure 4a) illustrated the existence of vanadium and oxygen in all the P-VO_2_, H-VO_2,_ and E-VO_2_ powders according to the V2p and O1s peaks. The V2p peaks were analyzed by Avantage of Thermo Scientific. In addition, Shirley background subtraction was employed to diminish the influence of heightened peaks due to signals from electrons undergoing inelastic scattering during the XPS characterization, ensuring a convincing quantification analysis of the peaks. As the high-resolution pattern for V2p shown in Figure 4b to d, both the V2p_3/2_ and V2p_1/2_ spectral peaks were significantly asymmetrical and each was split into two peaks, which implied that the vanadium of all the VO_2_ samples involved two different chemical states, corresponding to V^4+^ (The fitted purple curve) and V^5+^ (The fitted green curve), respectively. V^5+^ owing to V_2_O_5_ consisted of V2p_3/2_ at 517.5 eV and V2p_1/2_ at 525.0 eV. The gap between the V2p_3/2_ orbital and V2p_3/2_ orbital was 7.5 eV, which is in line with previous work [13]. Additionally, it is supposed that the existence of V^5+^ was caused by partial oxidation when the VO_2_ particles were exposed to air [36]. The V^4+^ attributed to VO_2_ was made up of V2p_3/2_ at 516.2 eV and V2p_1/2_ at 523.7 eV. In addition, the V2p_3/2_ of P-VO_2_ was most occupied with V^5+^, and V^4+^ only accounted for a small proportion. Comparably, the V^4+^ in the V2p orbitals of H-VO_2_ in Figure 4c showed the highest percentage, followed by E-VO_2_ in Figure 4d, and finally, the P-VO_2_ declared the least V^4+^ content in Figure 4b. Such results could be attributed to the surface lattice distortion of H-VO_2_ and E-VO_2_.

SEM was employed to investigate the morphology of the aforementioned VO_2_ particles. Due to the large size of the commercial precursor V_2_O_5_, the directly annealed P-VO_2_ particles inherited the large grain size, as shown in Figure 5a. It is evident that the particles aggregated severely to form large-scale clusters, with irregular shapes and different sizes ranging from 0.2 μm to 1.2 μm.Unfortunately, such sizes hindered P-VO_2_ from dispersing in the PVP matrix for film coating. Compared to the pristine sample, the degree of particle aggregation of quenched VO_2_ was much improved. In Figure 5b, the big clusters were broken into small H-VO_2_ parts with varying sizes, showing that the deionized water was able to split the large-size P-VO_2_ gathering during the quenching process. Especially, when the quenched solvent was replaced with ethanol with a smaller density, E-VO_2_ interacted more severely with the liquid and turned more entire separation into nanoparticles with approximately 200 nm in size, as presented in Figure 5c. In addition, the voids clearly appeared among the E-VO_2_ nanoparticles and the linkage knot hinted the E-VO_2_ separated from large-size P-VO_2_ particles. The observation that the quenched VO_2_ nanoparticles were easily separated into small pellets from large-sized clusters resulted from the thermodynamically unstable state of these aggregated clusters. This unstable state was due to the surface crystal distortion caused by the quenching process, confirming the results of XPS spectrums. It is worth mentioning that the particle was much larger than the value calculated in Figure 3a. The fact that the XRD resulted from Cu Kα radiation reflected crystalline particles rather than the actual morphology of the powders might explain the difference in the mean particle sizes. As for SEM, the signals of secondary electrons with much smaller De Broglie wavelengths were collected to reflect the topography of the particles. The different wavelengths of Cu Kα in XRD and secondary electrons in SEM caused different imaging results. On the other hand, VO_2_ particles in Figure 5 were assumed to be polycrystalline consisting of substantial crystals.

The DSC curves of the above P-VO_2_ H-VO_2_ and E-VO_2_ powders are illustrated in Figure 6a with expected exothermic and endothermic peaks. Such a thermal energy change suggested the phase transition behavior between VO_2_(M) and VO_2_(R). The exothermic peak (*T*_c,h_) at 68.35 °C and endothermic peak (*T*_c,c_) at 61.09 °C of P-VO_2_ implied the average *T*_c_ was 64.72 °C according to Equation (1). On the other hand, the energy involved in the MIT behavior was 47 J/g, approaching 51 J/g of the bulk VO_2_, showing good crystallinity of P-VO_2_ powders as shown in Figure 2. With the quenching process, the H-VO_2_ powders with phase transition peaks at 65.61 °C and 58.53 °C presented the average *T*_c_ as 62.07 °C, while the *T*_c_ of E-VO_2_ nanoparticles was calculated as 62.73 °C from the *T*_c,h_ peak at 65.73 °C and *T*_c,c_ peak at 58.24 °C. It is clear that the *T*_c_ of the quenched samples was slightly lowered, which could be attributed to the surface lattice distortion of the VO_2_ crystal structure as the XPS results suggested.

Considering the similar structure of H_2_O and C_2_H_6_O, FT−IR was employed to detect the -OH bond of the VO_2_ particles. In Figure 6b, the -OH bond corresponded to the peaks at 3450 cm^−1^ and hydrogen bonds appeared at 1630 cm^−1^. The integral area of the peak at 1630 cm^−1^ was used to reflect the relative content of the hydroxyl group in the samples. It turned out that the hydroxyl peak area of H-VO_2_ particles was the largest at 106.4, followed by the peak area of 87.7 of E-VO_2_, and finally the value of 25.3 of P-VO_2_. Such results indicated that the absorbed hydroxyl on the surface of both H-VO_2_ and E-VO_2_ particles was more than P-VO_2_. This could be due to the surface lattice distortion of the quenched samples, so there were more absorption sites bound to the hydroxyl group on the surface of these particles. Additionally, the peaks at around 990 cm^−1^ and 890 cm^−1^, respectively, corresponded to the stretching vibration and asymmetric stretching vibration of the V=O bond. The overlapping peaks at 720 cm^−1^ and 660 cm^−1^ were the characteristic peaks of VO_2_(M) [37]. In addition, the peaks at around 530 cm^−1^ were labeled as the stretching vibration of the V-O-V bond. Apart from the peak offset of the corresponding bonds in VO_2_, small extra peaks at around 1285 m^−1^ were discovered in H-VO_2_ and E-VO_2_, which was attributed to the O-H band after the quenching process. Such a phenomenon confirmed the surface lattice distortion to VO_2_ crystals of H-VO_2_ and E-VO_2_.

### 3.2. Thermochromic Properties and Morphology of VO_2_ Nanocomposite Films

To reduce the grain size of the as-synthesized particles, VO_2_ samples were mixed with PVP and ethanol in the mill tank to conduct thorough ball milling and centrifugation, ending up with a dark liquid mixture. The turbid upper solution was dried to collect the VO_2_-PVP compound, which was then mixed with additional ethanol to configure the spin-coating slurry. The slurry was then dropped on the soda-lime-silica glass substrate and spun to form the VO_2_ nanocomposite films. A UV3600 spectrophotometer coupled with a temperature-controlling device was served to characterize the transmittance of the films ranging from 300 nm to 2500 nm, and the thermochromic properties of the VO_2_ nanocomposite films were integrated by Equations (2) to (4). The solid lines in Figure 7a were tested at 20 °C, revealing the transmittance of M phase VO_2_ samples, and the dash lines were obtained at 90 °C, appearing the transmittance of R phase VO_2_ samples. As shown in Figure 7a and Table 1, the *T*_lum_ of P-VO_2_ film was 53.2% and Δ*T*_sol_ just reached 8.8%. The grain accumulation in Figure 5a was the cause for such poor properties. Comparably, the H-VO_2_ film sacrificed a small amount of luminous transmittance of 3.1% to achieve as enormous as 42.5% of improvement in solar-energy modulation ability to reach 12.5%. The reason for the enhanced Δ*T*_sol_ originated from the LSPR absorption peak of H-VO_2_ located at 1258 nm, which was stronger than the P-VO_2_ peak at 1293 nm. Such a phenomenon led to an enlarged gap in the transmittance of VO_2_ at 20 °C and 90 °C and contributed to higher Δ*T*_sol_. The E-VO_2_ nanocomposite film that came across with the same LSPR effect at 1150 nm revealed an exceeding increment of Δ*T*_sol_ of 72.2% (from 8.8% to 15.2%). In addition, the *T*_lum_ of E-VO_2_ film was boosted to 62.5%, indicating a better dispersity of the VO_2_ nanoparticles in the PVP matrix. The optimized luminous transmittance was also credited for less aggregation of the nanoparticles as shown in Figure 5c. In Figure 7b, the thermochromic performance of E-VO_2_ in this work was compared to previously reported VO_2_ nanocomposite films, and it was clear that the E-VO_2_ film exceeded most VO_2_ films [21,23,27,31,32,33]. Besides, the excellent Δ*T*_sol_ of the E-VO_2_ film was able to satisfy the demands of effectively regulating room temperature while the great *T*_lum_ met the requirement of indoor brightness, which was a great achievement for the potential application of VO_2_ thermochromic smart windows.

The morphology of the films was presented. It is clear that P-VO_2_ turned into nanoparticles with a size of around 100 nm after a ball milling process as described in Figure 8a. However, particle aggregation in Figure 5a was still common even though they were spin-coated to film. This aggregation structure was the reason for the unsatisfied thermochromic properties of the P-VO_2_ film. Comparably, grain accumulation also existed in H-VO_2_ film, which caused low luminous transmittance of the film. Additionally, cracks could be found in the film, and they were the cause of the inadequate performance of *T*_lum_. It is worth mentioning that most H-VO_2_ nanoparticles exhibited much smaller sizes than P-VO_2_ and they were dispersed individually in the PVP in Figure 8b, causing the LSPR effect and the improvement of Δ*T*_sol_. In Figure 8c, the E-VO_2_ nanoparticles showed an average size of tens of nanometers and nearly no sign of particle aggregation. Additionally, these particles with surface lattice distortion were highly isolated from each other and uniformly dispersed in the E-VO_2_ nanocomposite film. The large-scale surface morphology of E-VO_2_ film in Figure 8d introduced such dispersity of VO_2_ nanoparticles in a more intuitive perspective. This dispersion structure became beneficial for the optical properties of E-VO_2_ nanocomposite film.

## 4. Conclusions

As a strong electronic associated material, VO_2_(M) underwent a reversible phase transition between the monoclinic phase and rutile phase, corresponding with an abrupt change of near-infrared light transmittance, thus showing great potential in the smart window application. To improve the thermochromic properties of VO_2_, this paper applied a facile annealing method to synthesize VO_2_(M) powders. With the quenching treatment, the VO_2_ particles presented surface lattice distortion and they were individually dispersed in the PVP host to induce the LSPR effect, contributing to the exciting increment of solar-energy modulation ability. Hereby, the film corresponding to H-VO_2_ achieved an enormous improvement in Δ*T*_sol_ from 8.8% to 12.5%. Moreover, the film fabricated by the ethanol-quenched VO_2_ with no sign of aggregation showed an exceedingly high Δ*T*_sol_ of 15.2%, and the high dispersity of E-VO_2_ nanoparticles in the film contributed to much enhanced *T*_lum_ of 62.5%. Therefore, this work provided a new thought to promote the thermochromic performance of VO_2_ by particle quenching and the strategy was beneficial to the commercialization of VO_2_ thermochromic smart windows.

## Data Availability

Data are available on request from the authors.

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
