# Peer review of "Enhanced Thermochromic Performance of VO2 Nanoparticles by Quenching Process"

_nanomaterials, 2023, doi:10.3390/nano13152252_

Round 1

Reviewer 1 Report

The paper presents a comparative study of the structure and thermochromic properties (solar energy and luminous transmission modulation) for VO2 powders prepared by V2O5 reduction (with the process temperature optimization) followed by quenching in either water or ethanol. The obtained result indicating that quenching in ethanol provides significantly improved target performances is interesting and promising for the fabrication of commercial thermochromic devices such as “smart” windows. The paper can be recommended for publication upon some minor revisions, including:

- the abstract should include the most significant result relating to the comparison of quenching in ethanol and water;

- in the Equation 1 parameters ?c, ?c,h and ?c,c should be described;

- in order to properly compare the dispersion of the particles  SEM images  of H-VO2 and E-VO2 samples in Fig.7 should be presented in the same scale size;

- it would be better to discuss more clearly the correlation between the surface functional composition (particularly presence and content of hydroxyl and other groups on the surface) and maybe shape of the particles quenched in different media, their aggregation and thermochromic properties;

- some misprints should be corrected, e.g. “phase transition behavior between VO2(M) and VO2(M)” in line 262;

Minor improvement of English is required, e.g. in the Conclusion

"With quenching treatment, the VO2 particles were presented the individually dispersion state in the PVP host, contributing to the exciting increasement of luminous transmittance. And the induced LSPR effect gave rise to improvement of solar-energy modulation capability."  

Reviewer 2 Report

The manuscript “Enhanced Thermochromic Performance of VO2 Nanoparticles by Quenching Process” by S.Wu , L.Zhou , B.Li , S.Tian, X.Zhao  is devoted to fabrication, characterization and applications of  VO2 films with nanoparticles. The manuscript is well written, and the experimental data presented well describe and confirm all conclusions.  I think the article is interesting and could be accepted for publication with minor revision. 

Despite a well-written introduction for a wider range of readers, a number of questions remain closed.

- Since the article is essentially applied, at least hypothetical possibilities of exploring the obtained results should be discussed. 

- It remains completely unclear to what extent the proposed film fabricated by the ethanol quenched VO2 retain their properties after many heating and cooling cycles. Which, it would seem, is the key issue of technology?

- An equally important issue is the depth of transmission modulation. Despite the improvements presented in the article, its absolute value remains extremely small. The authors should discuss how much this is the limit and whether there is any benefit from the already achieved modulation value.

Reviewer 3 Report

Energy consumption has become an urgent issue not only for the global environment, but also for people’s lives.

To reduce energy consumption, new structures should be developed for glass surfaces to enhance their thermal insulation properties. Vanadium dioxide (VO2) is the most well-known thermochromic material, which exhibits a notable optical change from transparent to reflecting in the infrared upon a semiconductor-to-metal phase-transition

Although the research on VO2 smart window has been carried on for several decades, the real commercial use of it has not yet been achieved.

I consider it important to undertake research on this material, however, I do have a few comments:

Line 82-88: it would be good to write clearly here how much better the parameters of the materials obtained by the authors of this work are compared to the results obtained by other researchers. Of course you can find it in this article but it's better to make it more explicit. When it comes to structure, what kind of structure are we looking for.

Lin124: Eq. 1Eq illustrated.???

Line 129: Eq.1 - explain the meaning of the symbols

Fig1b: A more elaborate comment is needed because the diffraction spectra look very similar. On what basis, for example, better or worse crystallinity of the material is stated

Line 207-212: I understand that the size of the crystallites is based on the half widths of peak 28.5. It may be necessary to show how the half-width of this peak has changed in a separate Fig. or as an insert.

Fig. 4c and d show the deconvolution of the V2p peak. What program was used, how was the background determined?

Line 253-255: I note that the values obtained on the basis of XRD and TEM must be different because the observations were carried out using two different wavelengths, i.e. X-ray and electron

In general, the work is written in understandable language, but there are a few spelling mistakes, so I suggest you review it carefully

The work is written in understandable language but there are a few spelling mistakes so I suggest you review it carefully
